# Spherical Procrustes Alignment for Reliable Medical Audio Diagnosis

Ying Wang [1]  Guoheng Huang [2]  Chan-Tong Lam [1]  Xiaochen Yuan [1]

## Abstract

Reliable medical audio diagnosis requires models that are both accurate and honest about their uncertainty. However, fine-tuned models on small, imbalanced datasets often become overconfident due to norm bias, where predictions rely on feature magnitude rather than semantic alignment. The Equiangular Tight Frame (ETF) provides a theoretical optimum for class separation and is effective for imbalanced and calibration tasks due to its maximal angular separability and geometric fairness, but existing ETF-based methods perform poorly on noisy medical data because gradient rotation is unstable and fixed ETFs cannot adapt to drifting prototypes. To address this, we propose **S**pherical **P**rocrustes **A**lignment (SPA), which combines spherical constraints with dynamic ETF alignment. SPA uses a *Spherical branch* to eliminate norm bias via normalization and a *Geometric branch* to adapt features and align a fixed ETF with drifting prototypes via dynamic Procrustes alignment, while a *self-alignment mechanism* fuses the two branches to jointly optimize logits. Experiments on ICBHI 2017 and CirCor DigiScope show that SPA achieves state-of-the-art performance and transforms pre-trained models into reliable and efficient clinical tools without extra inference cost.

## 1. Introduction

Deep neural networks hold promise for medical audio diagnostics, such as respiratory sound (Rocha et al., 2018) and heart murmur classification (Oliveira et al., 2022). Yet, clinical adoption demands reliability (Guo et al., 2017); a high-confidence misclassification far outweighs the uncertainty of a 0.5 prediction (Lambert et al., 2024). Such

errors may mislead clinicians, resulting in delayed treatment or severe consequences. Therefore, diagnostic tools must achieve both high accuracy and calibration, ensuring confidence scores reflect true probabilities (Dong et al., 2025).

However, current models struggle to meet both goals (Niizumi et al., 2024; Fraihi et al., 2025; Jeong & Kim, 2025). This is due to the scarcity of medical audio data and extreme class imbalance (Rocha et al., 2018; Oliveira et al., 2022). This means that achieving high performance relies on transfer learning, which involves fine-tuning backbones (Kong et al., 2020a; Gong et al., 2021; Niizumi et al., 2021; Chen et al., 2023; Niizumi et al., 2024) pre-trained on large-scale datasets like ImageNet (Deng et al., 2009) or AudioSet (Gemmeke et al., 2017). While fine-tuning improves accuracy, it also causes overfitting and overconfidence. Severe imbalance worsens this issue, causing models to memorize normal samples and misclassify ambiguous ones with high confidence. *Figure 1* (c) illustrates this, showing that the Cross-Entropy (CE) baseline (gray) deviates severely from the diagonal, rendering the probabilities clinically useless.

To solve these overconfidence issues, existing solutions generally focus on two areas: training-time regularization (Lin et al., 2017; Cao et al., 2019; Szegedy et al., 2016; Zhang et al., 2018) and inference-time uncertainty quantification (Guo et al., 2017; Hasan et al., 2023; Lakshminarayanan et al., 2017). However, these methods, initially developed for general vision or audio tasks, face limitations given the sensitive nature of medical audio. Training-time strategies like Focal Loss (Lin et al., 2017; Cao et al., 2019) or Label Smoothing (Szegedy et al., 2016) operate at the loss level; they ignore feature geometry, causing miscalibration on hard samples. Similarly, Mixup (Zhang et al., 2018) creates non-clinical artificial patterns, leading to underfitting. Furthermore, inference-time methods introduce efficiency trade-offs. For example, Temperature Scaling (Guo et al., 2017) is merely a quick fix for logits and cannot address distorted representations. Sampling-based methods like Monte Carlo Dropout (Hasan et al., 2023) and Deep Ensembles (Lakshminarayanan et al., 2017) provide reliability but increase computational costs linearly, making them impractical for clinical use. Overall, these general solutions are ineffective because they treat the symptoms rather than the root cause of overconfidence in feature spaces: norm bias.

[1]Faculty of Applied Sciences, Macao Polytechnic University, Macao SAR, China [2]School of Computer Science and Technology, Guangdong University of Technology, Guangzhou, China. Correspondence to: Xiaochen Yuan <xcyuan@mpu.edu.mo>.

*Proceedings of the $43^{rd}$ International Conference on Machine Learning*, Seoul, South Korea. PMLR 306, 2026. Copyright 2026 by the author(s).

We trace the root cause of the overconfidence issue to a geometric pathology. The model conflates the magnitude of weights with semantic similarity (Wang et al., 2017). Under standard Cross-Entropy (CE), the logit $z \cdot w = \|z\|\|w\|\cos(\theta)$ is maximized not by improving semantic alignment $\cos(\theta)$ (Mukhoti et al., 2020). As shown in *Figure 1* (a), this creates a norm-biased radial space where decision boundaries are dominated by magnitude $\|z\|$ or $\|w\|$. Importantly, this pathology not only degrades calibration but also classification performance. The model mistakenly classifies high-norm noise or ambiguous background sounds as confident predictions simply because they fall into high-magnitude regions, overshadowing true semantic features (Wang et al., 2017). As a result, noisy or out-of-distribution (OOD) samples (marked by stars) trigger dangerously high confidence (e.g., 0.99). Therefore, we enforce a spherical constraint, setting both $\|z\|$ and $\|w\|$ to 1, as shown in *Figure 1* (b). Projecting features onto a unit hypersphere means that logits become pure cosine similarity, decoupling magnitude from confidence. Noisy samples yield low confidence values of 0.25.

However, enforcing only spherical constraints is insufficient. Medical audio features exhibit severe feature jitter, drifting unpredictably during training due to data scarcity and class imbalance. This instability poses a challenge for geometric regularization methods: fixed Equiangular Tight Frame (ETF) priors (Ni et al., 2025) fail to adapt to drifting prototypes, while gradient-based ETF rotation approaches (Gao et al., 2023) suffer from high variance and poor convergence on small, noisy medical datasets. This creates a critical issue, as existing geometric methods fail to eliminate norm bias and keep features from drifting in medical audio. To address this dual challenge, we propose a **S**pherical **P**rocrustes **A**lignment (SPA) method, the first one to align idealized geometric priors with the noisy reality of medical audio. It operates via a dual-branch decoupling strategy: The *Spherical branch* learns on a unit hypersphere to focus solely on semantic feature learning, eliminating norm bias. Meanwhile, the *Geometric branch* maintains a structurally perfect ETF to ensure geometric integrity. The core innovation bridging them is Dynamic Procrustes Alignment, which treats the alignment of ETF with drifting prototypes as an Orthogonal Procrustes Problem (Schönemann, 1966; Gower, 1975). By solving this via Singular Value Decomposition (SVD) (Golub & Reinsch, 1970), we compute the optimal rotation matrix that snaps the ETF structure to match the sphere's current prototypes in real time (*Figure 1* (b)), avoiding gradient instability. Finally, a Self-alignment mechanism transfers this aligned knowledge back to the sphere for joint optimization. As shown in *Figure 1* (c), while CE (gray) remains blindly overconfident, SPA (orange) aligns predictions closely with true accuracy (red), showing the effectiveness of our SPA. In summary, our contributions are:

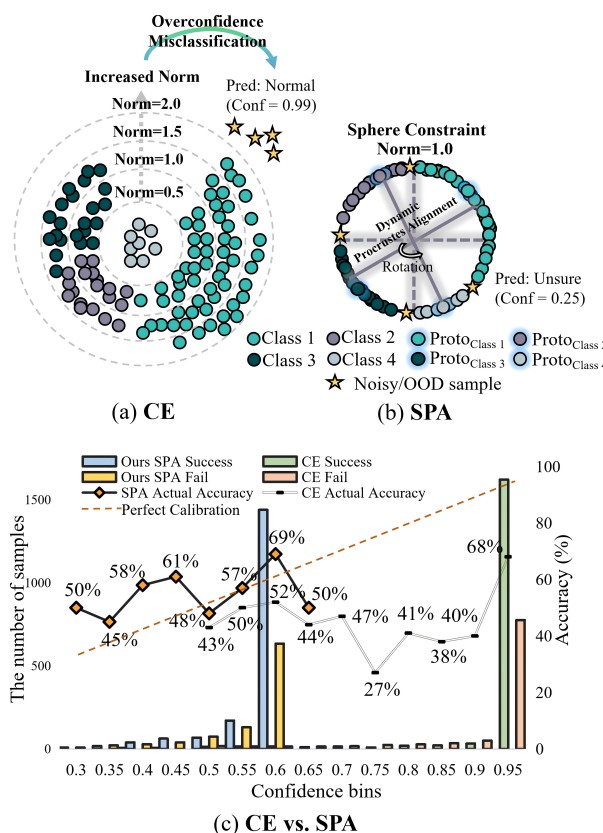

(a) **CE**       (b) **SPA**

(c) **CE vs. SPA**

*Figure 1.* Motivation and Effectiveness. (a) Standard CE training results in a norm-biased radial space: horizontal/vertical axes denote two feature dimensions, radial coordinate is $\|z\|$; high-magnitude noisy samples (stars) trigger overconfident prediction (0.99). (b) SPA enforces spherical constraint ($\|z\| = \|w\| = 1$), decoupling confidence from magnitude; Dynamic Procrustes Alignment adapts fixed ETF (gray skeleton) to drifting prototypes, producing uncertain prediction (0.25). (c) Reliability diagram on ICBHI: CE actual accuracy (gray squares) deviates from perfect calibration (red dashed); SPA actual accuracy (orange squares) aligns much closer to the diagonal, showing improved calibration.

- We identify norm bias, namely magnitude-driven overconfidence, as the geometric root of unreliable medical audio diagnosis. We also validate the finding that norm bias decouples confidence from clinical semantic alignment. This explains the calibration gap in fine-tuned medical audio models and shifts the solution paradigm from loss-level to geometric feature correction.

- We propose Spherical Procrustes Alignment (SPA), a dual-branch design tailored for medical audio that solves norm bias and feature jitter. SPA uses spherical projection to eliminate norm bias and Dynamic Procrustes Alignment to dynamically align ETF with drifting prototypes, ensuring stable geometry. Then, a self-alignment mechanism fuses the two branches to jointly optimize classification accuracy and calibration.

- Our SPA sets a new SOTA on the ICBHI and CirCor datasets, offering a reliable solution for clinical use. It provides high accuracy and well-calibrated confidence at no extra cost, turning over-parameterized backbones into trustworthy clinical tools.

## 2. Related Works

We review the landscape of medical audio diagnosis and analyze confidence calibration strategies. Finally, we discuss the geometric alignment theories that motivate our work.

**Medical Audio Diagnosis.** Medical audio diagnosis relies on fine-tuning pre-trained backbones that have evolved from convolutional neural networks (CNNs) (Kong et al., 2020b; Niizumi et al., 2021) to advanced Transformers, such as AST (Gong et al., 2021), BEATs (Chen et al., 2023), M2D (Niizumi et al., 2024), FBS (Fraihi et al., 2025), and patient-adaptive models like PAFA (Jeong & Kim, 2025). Despite their impressive diagnostic accuracy, these backbones often suffer from a calibration gap, wherein their predictive confidence fails to reflect the true clinical probability of a diagnosis. Most existing research treats these models as classification tools focused on maximizing label accuracy, ignoring the reliability of the output probability distribution (Arora et al., 2025). This causes these models to be blindly confident, giving high-probability scores to ambiguous or incorrectly classified samples. In contrast, our work not only pursues accuracy, but also ensures model calibration.

**Confidence Calibration.** Current efforts to perform calibration are generally divided into three types. First are regularization-based methods, such as Label Smoothing (Szegedy et al., 2016) and Mixup (Zhang et al., 2018), which soften training targets. While these methods are effective for general tasks, they often blur fine-grained acoustic features in medical audio, leading to underfitting. Second are post-hoc methods, such as Temperature Scaling (Guo et al., 2017), which rescale logits after training. Though computationally efficient, they only address the outputs and cannot fix distorted feature representations. Third are sampling-based methods, such as Deep Ensembles (Lakshminarayanan et al., 2017) and MC Dropout (Hasan et al., 2023), which provide robust uncertainty but require N inference passes, making them costly for devices with limited resources. Unlike these methods, our work solves the geometric root cause of overconfidence, enabling a single model to achieve good reliability without extra inference costs.

**Dynamic Alignment of Optimal Geometric Prototypes.** According to Neural Collapse (NC) theory (Papyan et al., 2020), the Simplex Equiangular Tight Frame (ETF) is the theoretically optimal geometry for class separation. Inspired by this, recent methods leverage the ETF for various tasks.

BalCAL (Ni et al., 2025) targets calibration by fusing a fixed ETF classifier with a learnable one. It uses a confidence-tunable module, adapter and dynamic adjustment, which address over- and underconfidence. RBL (Gao et al., 2023) solves long-tailed learning via gradient-based ETF rotation, using learnable orthogonal matrices and post-hoc logit adjustment to preserve ETF geometry. However, both are not suitable for medical audio. BalCAL's fixed ETF cannot adapt to drifting prototypes from scarce, imbalanced data, resulting in geometric misalignment. RBL's gradient-based rotation exhibits high variance and instability on small, noisy datasets and fails to address norm bias or calibration, both of which are critical for clinical reliability. To address this, our Spherical Procrustes Alignment (SPA) method uses a dual-branch design. The Spherical branch eliminates norm bias via unit hypersphere projection, while the Geometric branch uses Singular Value Decomposition (SVD)-solved Dynamic Procrustes Alignment to realign the fixed ETF with the drifting prototypes. Together, these branches stabilize feature geometry, eliminate jitter and achieve both norm bias elimination and calibration.

## 3. Methodology

### 3.1. Preliminaries

To ground the design of our Spherical Procrustes Alignment (SPA) method, we first formalize the geometric priors and optimization objectives for reliable medical audio diagnosis.

**Notations.** At the last layer of the pre-trained backbone, let $z \in \mathbb{R}^d$ denote the extracted medical audio feature vector, where d and K are the dimension of the feature vector and the number of diagnostic categories, respectively. The standard logit for class $k$ is:

$$Logit_k = W_k^T z = \|W_k\| \|z\| \cos \theta_k, \qquad (1)$$

where $\theta_k$ is the angular alignment between the feature $z$ and the classifier weight $W_k$, reflecting their semantic similarity.

**Ideal Geometric Prior: Simplex ETF.** We adopt the Simplex Equiangular Tight Frame (ETF) (Papyan et al., 2020) as the optimal geometric prior for class separation. A matrix $M \in \mathbb{R}^{d \times K}$ is a Simplex ETF if it satisfies:

$$M^T M = \frac{K}{K-1} \left( I_K - \frac{1}{K} \mathbf{1}_K \mathbf{1}_K^T \right), \qquad (2)$$

where $I_K$ is the $K \times K$ identity matrix, and $\mathbf{1}_K$ is the K-dimensional all-ones vector. The Simplex ETF is chosen for two advantages in medical audio diagnosis: 1) *Maximal Angular Separation*, which forces the model to learn discriminative features for all pathologies; and 2) *Geometric Fairness*, where prototype lengths are equal to keep majority classes from crowding out minority classes.

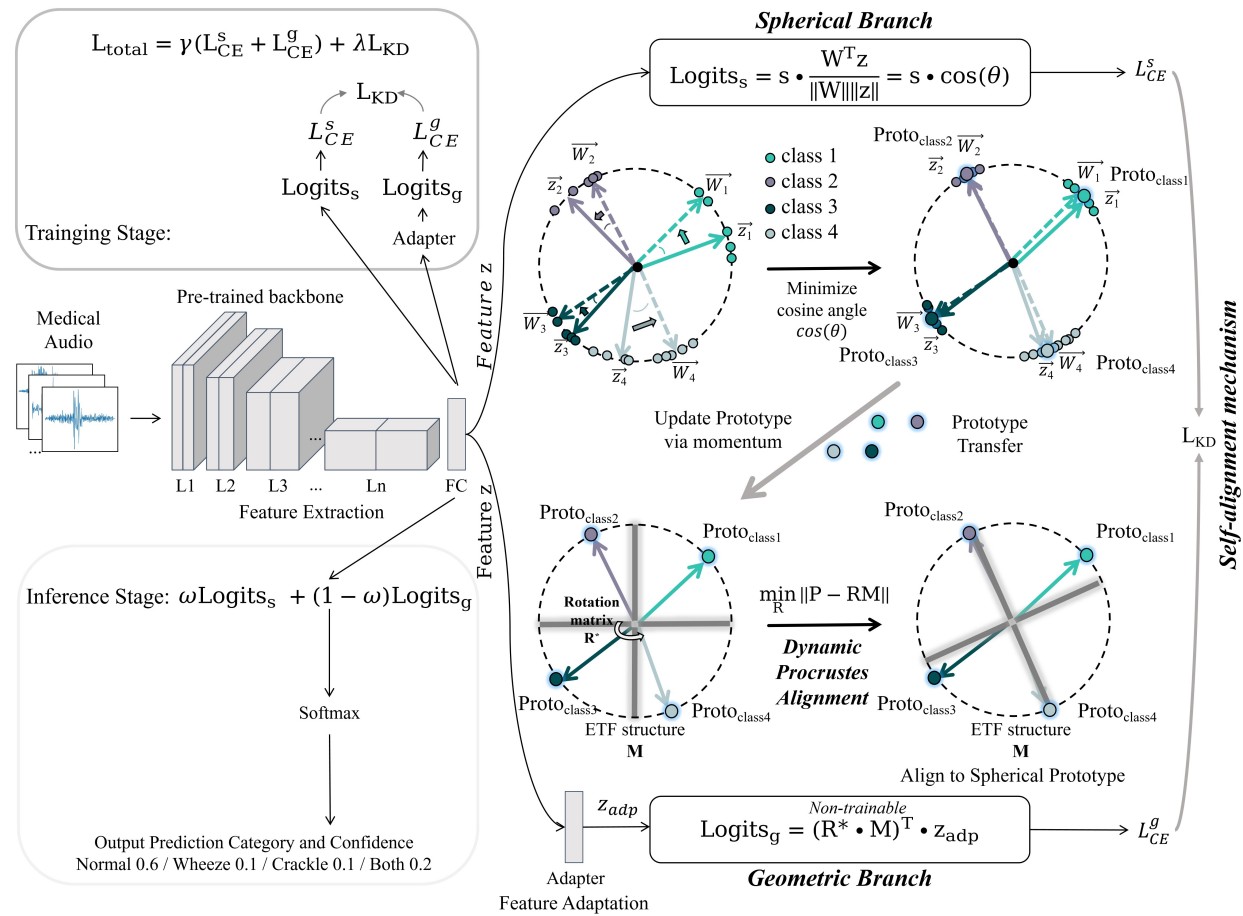

*Figure 2.* Overview of Spherical Procrustes Alignment (SPA). *Training stage*: Backbone extracts feature $z$; Spherical branch normalizes $z$ and $W$ to eliminate norm bias and updates prototypes; these prototypes are transferred to the Geometric branch, which aligns fixed ETF $M$ to the prototypes via Procrustes rotation $R^*$ to output stable logits; joint loss optimizes both branches. *Inference stage*: Fuse logits from both branches to output the diagnostic category and confidence.

**Problem Formulation.** The challenge of applying the Simplex ETF prior to pre-trained backbones for reliable medical audio diagnosis lies in meeting two optimization objectives: *Norm Decoupling*: Eliminate the prediction confidence reliance on the magnitudes of the features or weights ($\|z\|, \|W_k\|$), ensuring that confidence is solely determined by semantic angular alignment $\cos\theta_k$; *Dynamic Geometric Alignment*: Align the fixed Simplex ETF with drifting class prototypes caused by data scarcity and imbalance to stabilize geometric learning. To address both objectives, we propose SPA, a dual-branch design tailored for medical audio that decouples norm elimination and geometric stabilization. Its overall pipeline is illustrated in *Figure 2*.

### 3.2. Spherical Procrustes Alignment

As shown in *Figure 2*, our SPA takes the feature $z$ extracted by the pre-trained backbone as its input and operates via two synergistic branches: 1) To eliminate the norm bias, the **Spherical Branch** takes feature $z$ and classifier weights

$W$ as inputs. It then outputs calibrated logits $Logits_s$ after normalization, and updates class prototypes via momentum for transfer to the Geometric Branch; 2) To reference a perfect ETF classifier and dynamically align it with the prototypes, the **Geometric Branch** takes feature $z$ and the transferred prototypes as inputs. It generates an adapted feature $z_{adp}$ via an Adapter and computes the optimal rotation matrix $R^*$ by solving the Procrustes problem to align the ETF structure. It then outputs stable logits $Logits_g$; finally, the **Self-alignment mechanism** uses KL divergence to align the distributions of $Logits_s$ and $Logits_g$, and completes joint optimization with the cross-entropy losses of both branches. For inference, our SPA fuses $Logits_s$ and $Logits_g$ and outputs the final diagnostic confidence and category via Softmax.

**Spherical branch.** The Spherical branch computes calibrated logits by first performing $L_2$ normalization on both the feature vector $z$ and classifier weights $W_k$, projecting

them onto the unit hypersphere to yield $\tilde{z} = z/\|z\|_2$ and $\tilde{W}_k = W_k/\|W_k\|_2$. The classification logit is then redefined as a scaled cosine similarity:

$$Logits_{s,k} = s \cdot \tilde{W}_k^T \tilde{z} = s \cdot \cos(\theta_k) \quad (3)$$

where $s$ is a learnable scalar that scales the cosine value to a range suitable for the Softmax function. This process ensures that prediction confidence is strictly bounded by angular alignment $\theta_k$. Furthermore, when combined with the Geometric branch's ETF structure, an upper confidence bound for ambiguous samples can be derived in Appendix A.1.

The branch also updates class prototypes $P_k$ via momentum:

$$P_k \leftarrow mP_k + (1 - m)\mu_k \quad (4)$$

where $\mu_k$ is the mini-batch mean of normalized features $\tilde{z}$ for class $k$, and $m \in [0, 1)$ is the momentum coefficient.

**Geometric branch.** The Geometric branch computes stable logits by optimizing the *Dynamic Geometric Alignment* objective, leveraging a fixed Simplex ETF $M$ (gray skeleton in *Figure 2*) as a structural anchor. To avoid conflicts between backbone features and ETF geometry (Ni et al., 2025), we adopt a three-step process for dynamic alignment:

First, the feature $z$ is adapted via a learnable Adapter:

$$z_{adp} = A(z) = W_{adp}z + b_{adp} \quad (5)$$

where $W_{adp}$ and $b_{adp}$ are learnable parameters.

Next, the branch receives the momentum-updated prototypes $P$ from Eq. (4) and aligns the fixed ETF $M$ with $P$ by solving the Orthogonal Procrustes problem (Schönemann, 1966), the solution of which is in Appendix A.2:

$$\min_R \|P - RM\|_F^2 \quad \text{s.t.} \quad R^T R = I \quad (6)$$

The optimal rotation matrix $R^* = UV^T$ is computed via Singular Value Decomposition (SVD) (Golub & Reinsch, 1970) on the cross-correlation matrix $H = MP^T$ (where $U, \Sigma, V^T = \text{SVD}(H)$). This analytical solution avoids the gradient variance of learnable rotation matrices (Gao et al., 2023), with stability proven in Appendix A.3.

Finally, the branch computes its logits by projecting the adapted features onto the aligned ETF:

$$Logits_g = (R^*M)^T z_{adp} \quad (7)$$

**Self-alignment mechanism.** To bring the structural symmetry of the Geometric branch into the calibrated space of the Spherical branch, we couple the logits of the two branches, $Logits_s$ and $Logits_g$, via a joint loss function:

$$\mathcal{L}_{total} = \gamma(\mathcal{L}_{CE}^s + \mathcal{L}_{CE}^g) + \lambda\mathcal{L}_{KD}(Logits_s, Logits_g) \quad (8)$$

where $\mathcal{L}_{CE}$ is the Label Smoothing Cross-Entropy loss, $\mathcal{L}_{KD}$ is the KL-divergence loss scaled by a temperature parameter $\tau$, and $\gamma, \lambda$ are balancing coefficients.

### 3.3. Training and Inference Pipeline

The full training process is summarized in Algorithm 1, following an online update-then-align strategy. The data workflow is shown in Appendix C.1. In each mini-batch iteration, both branches first perform a forward pass to compute $Logits_s$ and $Logits_g$, then calculate the joint loss $\mathcal{L}_{total}$, which is used to update the network parameters via back-propagation. The Geometric branch then performs a gradient-free geometric update. It updates the prototypes $P$ with the current batch and immediately recalculates the rotation $R^*$ via SVD. This ensures that the branch provides an optimal geometric target for the next iteration.

---

**Algorithm 1** Training Pipeline of SPA

1: **Input:** Dataset $\mathcal{D}$, Pre-trained Backbone $f_\theta$, Fixed Simplex ETF $M$, Momentum coefficient $m$, Adapter $A_\phi$, Spherical classifier weights $W$, Initial learnable scale $s$, Balancing coefficients $\gamma, \lambda$, Temperature $\tau$.
2: **Output:** Optimized parameters $\theta, \phi, W, s$.
3: **Initialize:** Prototypes $P \leftarrow \mathbf{0}$, Rotation $R \leftarrow I$.
4: **for** epoch $t = 1$ **to** $T$ **do**
5:     **for** mini-batch $(x, y)$ in $\mathcal{D}$ **do**
6:         *// 1. Forward Pass & Parameter Update*
7:         Compute Spherical branch Logits using Eq. (3)
8:         Adapt feature using Eq. (5)
9:         Compute Geometric branch Logits using Eq. (7)
10:        Compute total Loss using Eq. (8)
11:        Update $\theta, W, \phi, s$ via optimizer.
12:        *// 2. Dynamic Procrustes Alignment*
13:        Compute batch prototype $\mu_k$
14:        Update Prototypes using Eq. (4)
15:        Align ETF to Prototypes via SVD (Eq. 6)
16:        Update rotation matrix $R \leftarrow R^*$
17:     **end for**
18: **end for**

---

During inference, we fuse both branches. $P_{final} = \omega\sigma(Logits_s) + (1 - \omega)\sigma(Logits_g)$, where $\sigma$ is the Softmax function. We set $\omega = 0.5$ in all experiments. The Spherical and Geometric branches work together to give reliable confidence at no extra cost.

## 4. Experiments

### 4.1. Experimental Setup

**Datasets and Evaluation Metrics.** We verify our SPA on two medical audio datasets: ICBHI 2017 (Rocha et al., 2018) and CirCor DigiScope (Oliveira et al., 2022). The

*Table 1.* **Performance on the ICBHI 4-Class Classification.** Best results are bolded.

| Backbone | Method | Classification Performance | | | Reliability |
|---|---|---|---|---|---|
| | | Se (%) ↑ | Sp (%) ↑ | AS (%) ↑ | ECE (%) ↓ |
| CNN6 | + CE Loss (Kong et al., 2020b) | $35.56_{\pm 4.99}$ | $81.75_{\pm 4.33}$ | $58.66_{\pm 0.54}$ | $23.53_{\pm 5.01}$ |
| | + Focal Loss (Lin et al., 2017) | $35.11_{\pm 5.36}$ | $79.18_{\pm 5.01}$ | $57.14_{\pm 0.58}$ | $17.51_{\pm 2.24}$ |
| | + LDAM-DRW (Cao et al., 2019) | $33.37_{\pm 3.96}$ | $80.49_{\pm 4.13}$ | $56.93_{\pm 0.85}$ | $17.47_{\pm 3.85}$ |
| | **+ SPA (Ours)** | $35.24_{\pm 2.92}$ | $82.96_{\pm 4.82}$ | $\mathbf{59.10}_{\pm 1.24}$ | $\mathbf{7.51}_{\pm 0.72}$ |
| AST | + CE Loss (Gong et al., 2021) | $42.58_{\pm 2.71}$ | $79.52_{\pm 2.07}$ | $61.05_{\pm 0.54}$ | $21.59_{\pm 6.99}$ |
| | FBS (Fraihi et al., 2025) | $43.22_{\pm 2.44}$ | $84.19_{\pm 3.08}$ | $64.28_{\pm 1.09}$ | $19.82_{\pm 5.13}$ |
| | **FBS + SPA (Ours)** | $44.81_{\pm 2.65}$ | $84.92_{\pm 3.19}$ | $\mathbf{65.01}_{\pm 2.77}$ | $\mathbf{5.49}_{\pm 1.13}$ |
| BEATs | + CE Loss (Chen et al., 2023) | $50.60_{\pm 1.79}$ | $77.20_{\pm 3.22}$ | $63.90_{\pm 1.15}$ | $28.51_{\pm 4.91}$ |
| | **+ SPA (Ours)** | $49.77_{\pm 2.17}$ | $79.06_{\pm 2.81}$ | $\mathbf{64.42}_{\pm 0.78}$ | $\mathbf{4.44}_{\pm 1.02}$ |
| | PAFA (Jeong & Kim, 2025) | $48.72_{\pm 3.75}$ | $80.19_{\pm 4.07}$ | $64.45_{\pm 0.52}$ | $23.36_{\pm 7.83}$ |
| | **PAFA + SPA (Ours)** | $49.34_{\pm 4.12}$ | $81.21_{\pm 4.48}$ | $\mathbf{65.27}_{\pm 0.57}$ | $\mathbf{6.05}_{\pm 3.63}$ |

ICBHI dataset contains 6,898 respiratory cycles with a severe class imbalance ratio of 11:1. Following the official 60/40 split, we test on 4-class task and report Sensitivity (Se), Specificity (Sp), Average Score (AS), and Expected Calibration Error (ECE) (Guo et al., 2017). The CirCor dataset includes 5,272 phonocardiograms for heart murmur detection. Following the split ratio in (Niizumi et al., 2024), we report Weighted Accuracy ($W_{acc}$), Unweighted Average Recall (UAR), and ECE.

**Baseline Methods.** We benchmark SPA against a standard fine-tuning baseline (Cross-Entropy (CE)) and three categories of advanced methods: *1) Imbalanced Learning:* Focal Loss (Lin et al., 2017) and LDAM-DRW (Cao et al., 2019). *2) Confidence Calibration:* Label Smoothing (Szegedy et al., 2016), Mixup (Zhang et al., 2018), Temperature Scaling (Guo et al., 2017), Monte-Carlo (MC) Dropout (Lemay et al., 2022), and Deep Ensembles (Lakshminarayanan et al., 2017). *3) Audio Extraction Backbones:* CNN6, CNN14 (Kong et al., 2020a), AST (Gong et al., 2021), BEATs (Chen et al., 2023), BYOL-A (Niizumi et al., 2021), and M2D (Niizumi et al., 2024).

**Implementation Details.** For the ICBHI dataset, we follow the preprocessing way of Jeong et al. (Jeong & Kim, 2025); for the CirCor dataset, we follow the settings of Niizumi et al. (Niizumi et al., 2024). To adapt to different pre-trained backbones, SPA uses backbone-specific optimal hyperparameters. For PAFA, it uses $\tau = 2.0, \lambda = 0.5$, and $m = 0.99$. For M2D, it uses $\tau = 4.0, \lambda = 0.5$, and $m = 0.95$. All results report the mean and standard deviation over 5 seeds (ICBHI) or 3 seeds (CirCor). Other details are provided in Appendix B.1 and B.2.

### 4.2. Comparison with state-of-the-art methods

**Performance on the ICBHI 2017 dataset.** We first verify the effectiveness of our SPA on the ICBHI dataset by

*Table 2.* **Confidence Calibration Comparison on BEATs Backbone.** Inf. Time is the average latency (ms) per batch (size 32) on an NVIDIA RTX 8000 GPU. Best results are bolded.

| Method | AS (%) ↑ | ECE (%) ↓ | Inf. Time (ms) |
|---|---|---|---|
| CE Loss | $63.90_{\pm 1.15}$ | $28.51_{\pm 4.91}$ | 4.1 |
| Label Smoothing | $62.58_{\pm 0.59}$ | $33.91_{\pm 4.41}$ | 4.1 |
| Mixup | $62.16_{\pm 0.66}$ | $8.67_{\pm 6.29}$ | 4.1 |
| *Post-hoc & Sampling Methods* | | | |
| Temperature Scaling | $63.90_{\pm 1.15}$ | $21.99_{\pm 6.97}$ | 4.1 |
| MC Dropout (5 passes) | $63.82_{\pm 0.89}$ | $25.90_{\pm 4.18}$ | 19.1 |
| Deep Ensembles (5 models) | $64.40_{\pm 0.93}$ | $22.29_{\pm 4.89}$ | 20.5 |
| **SPA (Ours)** | $\mathbf{64.42}_{\pm 0.78}$ | $\mathbf{4.44}_{\pm 1.02}$ | **4.2** |

comparing it with imbalanced learning methods and strong audio baselines, as shown in Table 1. Imbalanced learning methods (Focal Loss (Lin et al., 2017) and LDAM-DRW (Cao et al., 2019)) fail to improve AS or reduce ECE, as they operate only at the loss level without rectifying the distorted feature geometry. In contrast, SPA restructures the feature space. Applying SPA to the CNN6 (Kong et al., 2020a) backbone increases AS to 59.10% and reduces ECE to 7.51%. This trend holds for stronger backbones as well. While BEATs (Chen et al., 2023) achieves high accuracy, it suffers from extreme overconfidence (ECE 28.51%). SPA calibrates BEATs, reducing ECE to 4.44% while increasing AS to 64.42%. Furthermore, Integrating SPA with PAFA achieves a State-Of-The-Art (SOTA) AS of 65.27%, confirming that solving the geometric pathology is a universal booster for various backbones.

**Comparison with Confidence Calibration Techniques.** We compare SPA with existing calibration methods in Table 2 using the BEATs backbone on ICBHI dataset. Regularization tools during training cannot balance accuracy and reliability. Label Smoothing (Szegedy et al., 2016) gives a high ECE of 33.91% while Mixup (Zhang et al., 2018) loses accuracy, dropping AS to 62.16%. Post-hoc scaling (Guo et al., 2017) only lowers ECE to 21.99% without fix-

*Table 3.* **Performance on CirCor Dataset for heart murmur detection.** Best results are bolded. "-" denotes the unreported results.

| Backbone | Method | Classification | | Reliability |
|---|---|---|---|---|
| | | $W_{acc}$ (%) ↑ | UAR (%) ↑ | ECE (%) ↓ |
| *Previous Studies* | | | | |
| Wav2vec | Panah et al. (Panah et al., 2023) | 80.0 | 70.0 | - |
| HSMM | CUED Acoustics (McDonald et al., 2022) | 80.0 | 68.0 | - |
| *Pre-trained extractors* | | | | |
| CNN14 | + CE Loss (Kong et al., 2020a) | $57.47_{\pm3.25}$ | $53.63_{\pm2.89}$ | $11.24_{\pm2.14}$ |
| | **+ SPA (Ours)** | $\mathbf{58.32}_{\pm2.47}$ | $\mathbf{54.47}_{\pm2.18}$ | $\mathbf{6.08}_{\pm1.24}$ |
| BYOL-A | + CE Loss (Niizumi et al., 2021) | $54.34_{\pm3.71}$ | $54.81_{\pm2.67}$ | $12.13_{\pm2.53}$ |
| | **+ SPA (Ours)** | $\mathbf{55.73}_{\pm2.85}$ | $\mathbf{55.52}_{\pm2.36}$ | $\mathbf{6.42}_{\pm1.47}$ |
| AST | + CE Loss (Gong et al., 2021) | $64.12_{\pm2.38}$ | $66.38_{\pm2.95}$ | $10.35_{\pm1.87}$ |
| | **+ SPA (Ours)** | $\mathbf{65.24}_{\pm1.96}$ | $\mathbf{66.91}_{\pm2.07}$ | $\mathbf{5.58}_{\pm1.12}$ |
| M2D | + CE Loss (Niizumi et al., 2024) | $83.51_{\pm1.92}$ | $72.14_{\pm2.14}$ | $9.17_{\pm1.65}$ |
| | **+ SPA (Ours)** | $\mathbf{84.23}_{\pm1.53}$ | $\mathbf{72.96}_{\pm1.78}$ | $\mathbf{4.63}_{\pm0.89}$ |

ing the distorted feature space. Sampling methods like MC Dropout (Lemay et al., 2022) and Deep Ensembles (Lakshminarayanan et al., 2017) are very slow. They increase inference time to 19.1 ms and 20.5 ms, which is about five times the baseline cost. In contrast, SPA achieves the highest AS of 64.42% and the best calibration with an ECE of 4.44%. Most importantly, SPA keeps a fast inference time of 4.2 ms. This is nearly the same as the 4.1 ms baseline. These results show SPA provides accurate and reliable diagnoses with almost no extra time cost.

**Performance on the CirCor DigiScope dataset.** We further assess the generalization capability of our SPA on the CirCor dataset for heart murmur detection, applying it across four diverse pre-trained backbones, as shown in Table 3. Our SPA improves $W_{acc}$ and reduces ECE by approximately 50% across all supervised (CNN14 (Kong et al., 2020a) and AST (Gong et al., 2021)) and self-supervised (BYOL-A (Niizumi et al., 2021) and M2D (Niizumi et al., 2024)) backbones. Notably, for the M2D backbone, SPA achieves a new SOTA $W_{acc}$ of 84.23% with a minimal ECE of 4.63%. These results indicate that the benefits of spherical constraints and geometric alignment are robust to different data distributions and architectures.

### 4.3. Ablation Studies

**The Effectiveness of Each Component.** To validate the individual contributions of each component, we perform an ablation study on the ICBHI dataset using the BEATs backbone, as shown in Table 4. The baseline BEATs model exhibits severe overconfidence (ECE 28.51%) due to norm inflation. Applying the Spherical branch (Sph.) alone eliminates this magnitude bias, reducing ECE to 19.37% and confirming our Norm Bias hypothesis. Introducing the Fixed ETF geometry forces maximal class separation, further lowering ECE to 9.37%, though the rigid target causes a slight

drop in AS. Adding Dynamic Procrustes Alignment (Proc.) rotates the fixed ETF to match the sphere, recovering the lost accuracy (63.48%) and reducing ECE to 4.87%. Finally, the Self-distillation mechanism yields the optimal trade-off, proving that all components are vital.

*Table 4.* **Component analysis for ICBHI dataset based on the BEATs backbone.**

| Components | | | | Performance | |
|---|---|---|---|---|---|
| Sph. | ETF | Proc. | Self-Dist. | AS (%) ↑ | ECE (%) ↓ |
| × | × | × | × | $63.90_{\pm1.15}$ | $28.51_{\pm4.91}$ |
| ✓ | × | × | × | $63.96_{\pm1.02}$ | $19.37_{\pm6.60}$ |
| ✓ | ✓ | × | × | $62.83_{\pm0.92}$ | $9.37_{\pm2.11}$ |
| ✓ | ✓ | ✓ | × | $63.48_{\pm0.74}$ | $4.87_{\pm1.42}$ |
| ✓ | ✓ | ✓ | ✓ | $\mathbf{64.42}_{\pm0.78}$ | $\mathbf{4.44}_{\pm1.02}$ |

**Hyperparameter Sensitivity Analysis.** Table 5 illustrates the impact of hyperparameters on SPA performance via the control variable method. Two parameters are fixed at their optimal values while the third is changed. The weight $\lambda$ performs best at 0.5. Lower or higher values lead to weak regularization or overfitting. High momentum ($m = 0.99$) is vital for stability. Without momentum ($m = 0.0$), the ECE increases to 7.2% because noisy batches disrupt the geometric alignment. Finally, a temperature of 2.0 helps to smooth the probability output. If the temperature is too low, the calibration error increases. These results show that our settings balance diagnostic accuracy and reliability.

### 4.4. Visualization Results and Analysis

**Rectifying Geometric Pathology.** Standard training on imbalanced data often results in a distorted feature space where majority classes compress minority ones. We visualize this pathology and SPA in *Figure 3*. In *Figure 3* (a), the SOTA method PAFA shows irregular inter-class cor-

*Table 5.* **Sensitivity Analysis of SPA Hyperparameters on ICBHI.** We use a control variable method where two parameters are fixed at optimal values while the third varies. Best results are Bolded.

| Parameter | Value | AS (%) ↑ | ECE (%) ↓ |
|---|---|---|---|
| Alignment Weight ($\lambda$) | 0.1 | $61.8_{\pm0.95}$ | $8.2_{\pm1.20}$ |
| | **0.5** | $\mathbf{64.4_{\pm0.78}}$ | $\mathbf{4.4_{\pm1.02}}$ |
| | 1.0 | $63.2_{\pm0.82}$ | $6.1_{\pm1.15}$ |
| Momentum ($m$) | 0.0 | $62.1_{\pm1.12}$ | $7.2_{\pm1.45}$ |
| | 0.9 | $63.5_{\pm0.88}$ | $5.3_{\pm1.08}$ |
| | **0.99** | $\mathbf{64.4_{\pm0.78}}$ | $\mathbf{4.4_{\pm1.02}}$ |
| Temperature ($\tau$) | 1.0 | $63.1_{\pm0.75}$ | $6.8_{\pm0.98}$ |
| | **2.0** | $\mathbf{64.4_{\pm0.78}}$ | $\mathbf{4.4_{\pm1.02}}$ |
| | 4.0 | $64.0_{\pm0.91}$ | $5.1_{\pm1.12}$ |

relations (-0.26 to 0.01), indicating that the Normal class geometrically dominates the space. In contrast, SPA enforces geometric fairness, with off-diagonal correlations uniform at approximately 0.0. The t-SNE projections in *Figure 3* (c) further evidence this structural optimization. While PAFA exhibits blurred boundaries and overlapping regions, SPA produces compact, well-separated clusters with distinct angular margins. The features collapse around their respective prototypes, creating clear angular voids that prevent ambiguous samples from triggering high confidence. This rectification reduces minority suppression, increasing true positives for the Wheeze class (from 154 to 160) without sacrificing majority performance, as shown in *Figure 3* (b).

**Calibration by Decoupling Magnitude.** To show that the spherical branch eliminates norm bias, we analyze the reliability diagrams across different backbones in the CirCor dataset, as shown in *Figure 4*. Baseline backbones (red curves) sag below the diagonal, which is a sign of overconfidence caused by unchecked magnitude inflation in the logits. Our SPA (blue curves) alters this behavior by constraining predictions to the unit hypersphere. This forces the model to rely solely on angular semantics rather than norm length, pulling the calibration curves to the diagonal. For the M2D backbone in *Figure 4* (d), SPA achieves near-perfect alignment (ECE 4.63%), showing that decoupling magnitude is a backbone-agnostic solution for reliable clinical diagnosis.

## 5. Conclusion and Future Work

We attribute unreliability in medical audio to norm bias, which inflates confidence in semantic alignment. Our Spherical Procrustes Alignment (SPA) method decouples geometry from representation. It uses spherical constraints and dynamic alignment to bridge idealized priors with noisy features. This transforms backbones into calibrated clinical tools that achieve state-of-the-art accuracy and reliability on ICBHI and CirCor at no extra costs. Our work shows that geometric integrity is as important as the power of feature

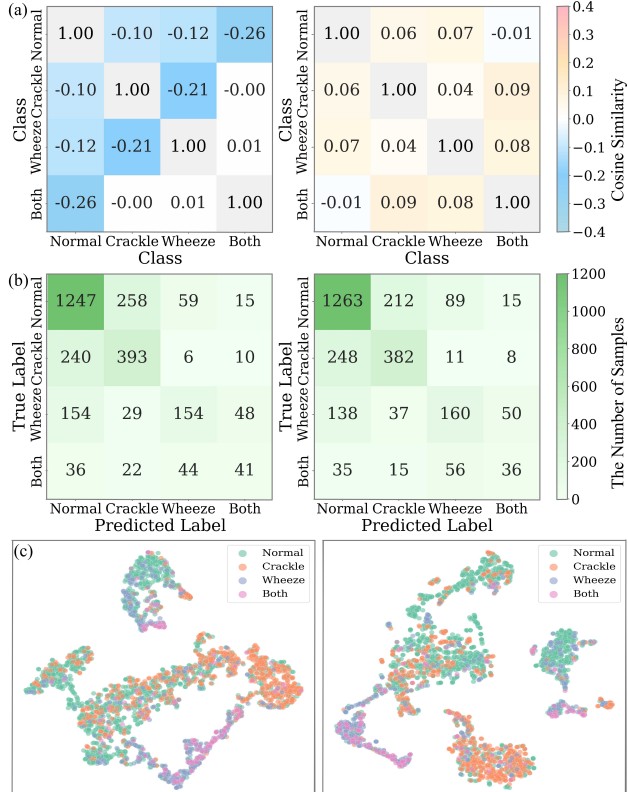

*Figure 3.* Qualitative comparison between the SOTA method PAFA and Ours (PAFA + SPA) on the ICBHI dataset: (a) Cosine Similarity Heatmaps of class prototypes, (b) Confusion Matrices, and (c) t-SNE feature projections.

extraction in data-scarce medical tasks.

In our SPA, the momentum-based prototype update assumes clean labels. High label noise can cause the prototypes to deviate from the true class centre. Future work will introduce robust prototype estimation with outlier rejection. Furthermore, future work will use angular voids between prototypes to detect out-of-distribution data reliably. Samples in these empty spaces are likely to represent noise or unknown pathologies, providing a more reliable signal than standard probability scores. Our goal is also to apply the SPA as a geometric adapter for audio foundation models. This will help stabilize feature geometry during fine-tuning and enable large models to adapt to small medical datasets without overfitting.

## Impact Statement

Our work make medical audio diagnosis more reliable. In clinical settings, a model that is overconfident about a wrong answer can mislead doctors and harm patients. By solving the geometric root of overconfidence, our method ensures that AI confidence scores reflect true probabilities. This

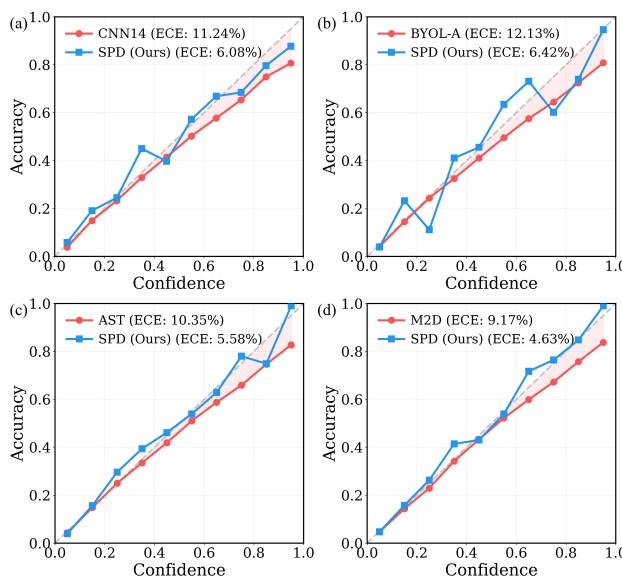

*Figure 4.* Reliability diagrams on the CirCor dataset across four backbones: (a) CNN14, (b) BYOL-A, (c) AST, and (d) M2D.

improves the safety of using AI for respiratory and heart sound screening. Beyond safety, this technology can help regions with few medical specialists. Accurate and calibrated diagnostic tools allow for earlier detection of diseases in resource-poor areas.

## Acknowledgements

This work was supported by the Macao Polytechnic University under grant RP/FCA-12/2025 and the Science and Technology Development Fund of Macau SAR under Grant 0053/2025/RIB2.

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

# A. Theoretical Analysis

## A.1. Theorem 1. Confidence Upper Bound for OOD or noisy Samples

**Theorem.** Assume $\|z\|_2 = 1$ and $\|W_k\|_2 = 1$ for all classes, let $\text{logit}_k = sW_k^\top z = s\cos\theta_k$, and define

$$p_k = \frac{\exp(s\cos\theta_k)}{\sum_{j=1}^{K} \exp(s\cos\theta_j)}. \tag{9}$$

If the feature vector satisfies $\cos\theta_k \leq \cos(\theta_{\min} + \delta)$ for every class, where $\theta_{\min} = \arccos(-1/(K-1))$ and $\delta > 0$, then

$$\max_k p_k \leq \frac{1}{1 + (K-1)\exp\left(-\frac{sK}{K-1}\cos(\theta_{\min} + \delta)\right)}. \tag{10}$$

**Proof.** Let $c_k := \cos\theta_k$ and denote $k^* := \arg\max_k c_k$. Then the maximum softmax probability is:

$$\max_k p_k = \frac{\exp(sc_{k^*})}{\sum_{j=1}^{K} \exp(sc_j)}. \tag{11}$$

Split the denominator:

$$\max_k p_k = \frac{\exp(sc_{k^*})}{\exp(sc_{k^*}) + \sum_{j\neq k^*} \exp(sc_j)}. \tag{12}$$

Apply Jensen's inequality to the convex function $\exp(\cdot)$ for the $K-1$ terms:

$$\frac{1}{K-1}\sum_{j\neq k^*} \exp(sc_j) \geq \exp\left(\frac{s}{K-1}\sum_{j\neq k^*} c_j\right). \tag{13}$$

Multiplying both sides by $K-1$:

$$\sum_{j\neq k^*} \exp(sc_j) \geq (K-1)\exp\left(\frac{s}{K-1}\sum_{j\neq k^*} c_j\right). \tag{14}$$

For a simplex ETF, the classifier weights satisfy $\sum_{j=1}^{K} W_j = 0$. Taking inner product with $z$:

$$\sum_{j=1}^{K} c_j = \sum_{j=1}^{K} W_j^\top z = \left(\sum_{j=1}^{K} W_j\right)^\top z = 0^\top z = 0. \tag{15}$$

Therefore,

$$\sum_{j\neq k^*} c_j = -c_{k^*}. \tag{16}$$

Substitute Eq. (16) into Eq. (14):

$$\sum_{j\neq k^*} \exp(sc_j) \geq (K-1)\exp\left(-\frac{sc_{k^*}}{K-1}\right). \tag{17}$$

Plug this lower bound into the denominator of Eq. (12):

$$\max_k p_k \leq \frac{\exp(sc_{k^*})}{\exp(sc_{k^*}) + (K-1)\exp\left(-\frac{sc_{k^*}}{K-1}\right)}. \tag{18}$$

Factor $\exp(sc_{k^*})$ from the denominator:

$$\max_k p_k \leq \frac{\exp(sc_{k^*})}{\exp(sc_{k^*})\left[1 + (K-1)\exp\left(-\frac{sc_{k^*}}{K-1} - sc_{k^*}\right)\right]}. \tag{19}$$

Cancel $\exp(sc_{k^*})$ and simplify the exponent:

$$-\frac{sc_{k^*}}{K-1} - sc_{k^*} = -sc_{k^*}\left(\frac{1}{K-1} + 1\right) = -\frac{sK}{K-1}c_{k^*}. \tag{20}$$

Thus,

$$\max_k p_k \leq \frac{1}{1 + (K-1)\exp\left(-\frac{sK}{K-1}c_{k^*}\right)}. \tag{21}$$

Define $f(c) = \frac{1}{1+(K-1)e^{-\frac{sK}{K-1}c}}$. Since the exponential function is decreasing, $f(c)$ is increasing in $c$. By the theorem's assumption, $c_{k^*} \leq \cos(\theta_{\min} + \delta)$. Therefore,

$$f(c_{k^*}) \leq f(\cos(\theta_{\min} + \delta)). \tag{22}$$

Substituting this into Eq. (21) gives the desired bound:

$$\max_k p_k \leq \frac{1}{1 + (K-1)\exp\left(-\frac{sK}{K-1}\cos(\theta_{\min} + \delta)\right)}. \tag{23}$$

As $\delta$ increases, $\cos(\theta_{\min}+\delta)$ decreases, the exponential term $\exp\left(-\frac{sK}{K-1}\cos(\theta_{\min} + \delta)\right)$ grows, the denominator increases, and the upper bound approaches $1/K$, forcing near-uniform confidence for noisy or OOD samples.

## A.2. Solution to the Orthogonal Procrustes Problem

For completeness, we present the derivation of the solution to the Orthogonal Procrustes problem. This is a standard result in linear algebra. The objective is to find an orthogonal matrix $R$ that minimizes:

$$\min_R \|P - RM\|_F^2 \quad \text{s.t.} \quad R^T R = I. \tag{24}$$

**Derivation:** Expanding the Frobenius norm:

$$\|P - RM\|_F^2 = \text{Tr}((P - RM)^T(P - RM)) \tag{25}$$
$$= \text{Tr}(P^T P - P^T RM - M^T R^T P + M^T M). \tag{26}$$

The terms $\text{Tr}(P^T P)$ and $\text{Tr}(M^T M)$ are constant. Thus, the minimization problem is equivalent to:

$$\max_R \text{Tr}(P^T RM). \tag{27}$$

Using the cyclic property of the trace, $\text{Tr}(P^T RM) = \text{Tr}(RMP^T) = \text{Tr}(RH^T)$, where $H = MP^T$. Performing SVD on $H$ yields $H = U\Sigma V^T$. Substituting, we get:

$$\text{Tr}(RH^T) = \text{Tr}(RV\Sigma U^T) = \text{Tr}(U^T RV\Sigma). \tag{28}$$

Let $Z = U^T RV$. Since $U$, $R$, and $V$ are orthogonal, $Z$ is also orthogonal. The problem reduces to $\max_Z \text{Tr}(Z\Sigma)$, which is maximized when $Z = I$. This leads to the solution:

$$R^* = UV^T. \tag{29}$$

This solution is efficiently computable and is core to our dynamic alignment procedure.

### A.3. Alignment Stability Analysis

**Theorem 2.** *The integration of Momentum Prototyping and analytical SVD alignment (Eq. 4, 6) strictly reduces the variance of the geometric target compared to gradient-based rotation updates, eliminating geometric jitter in low-data regimes.*

**Proof & Analysis:** Let the ideal class center be $\mu^*$ and the current mini-batch mean be $\mu_B = \mu^* + \xi$, where $\xi \sim \mathcal{N}(0, \sigma^2 I)$ represents the sampling noise caused by data scarcity.

**1. Instability of Gradient-based Rotation in RBL (Gao et al., 2023)** In gradient-based methods, the rotation matrix $R$ is updated via SGD: $R_{t+1} \leftarrow R_t - \eta \nabla_R \mathcal{L}$. The gradient $\nabla_R \mathcal{L}$ attempts to align the ETF with the noisy batch $\mu_B$.

$$\nabla_R \mathcal{L} \propto (\mu^* + \xi) M^T. \tag{30}$$

Here, the noise term $\xi M^T$ directly injects high-variance disturbances into the rotation update. Since rotation matrices lie on a manifold (Lie Group $SO(d)$), this noise causes $R_t$ to oscillate randomly on the tangent space. We term this phenomenon **geometric jitter**. The spherical branch must effectively chase a moving target, preventing the features from tightly collapsing.

**2. Stability of SPA** In our SPA method, we decouple the accumulation of statistics from the alignment calculation. First, the prototype $P$ is updated via momentum $m$:

$$P_t = m P_{t-1} + (1 - m)(\mu^* + \xi_t) = \sum_{i=0}^{t} (1 - m) m^{t-i} (\mu^* + \xi_i). \tag{31}$$

This acts as an exponential moving average. The variance of the noise in $P_t$ is reduced significantly:

$$\text{Var}(P_t) \approx \frac{1 - m}{1 + m} \sigma^2 \ll \sigma^2. \tag{32}$$

When $m \to 1$, the noise is suppressed by orders of magnitude. Second, the SVD operation ($R^* = VU^T$) computes the *global optimum* for this stable $P_t$ instantaneously. Unlike SGD, which takes a small, noisy step $\eta$, SVD snaps the geometry to the most stable estimation available.

**3. Geometric Interpretation: Radial vs. Tangential Forces** The stability of $R^*$ changes the nature of the gradient received by the spherical backbone. The loss gradient w.r.t the feature $z$ is:

$$\nabla_z \mathcal{L} \propto z - R^* m_k. \tag{33}$$

- **With Jitter (Gradient $R$):** If $R$ oscillates, the target $R_k^m$ moves tangentially relative to the hypersphere surface. The spherical branch wastes optimization energy chasing this tangential shift ($\nabla_{tan}$), leading to slow convergence.

- **With Stability (SPA $R^*$):** Since $R^*$ is stable, the tangential component is minimized ($\nabla_{tan} \approx 0$). The gradient force becomes purely **radial** ($\nabla_{rad}$), pushing the feature $z$ directly toward its assigned angular slot.

This confirms that SPA ensures the model focuses solely on semantic compression rather than compensating for geometric misalignment.

## B. Experiments

### B.1. Experimental Environment

Experiments were conducted on a cluster with $8 \times$ NVIDIA Quadro RTX 8000 GPUs (48 GB VRAM), dual Intel Xeon CPUs (256 GB RAM), and Ubuntu 24.04. The software stack includes Python 3.10.16, PyTorch 2.6.0, CUDA 12.4, and Librosa 0.10.2. Models were evaluated using random seeds [1, 2, 3, 4, 5] (ICBHI) and three stratified splits (CirCor).

### B.2. Detailed Implementation Settings

**Audio Preprocessing.** To maintain consistency across respiratory (ICBHI) and heart sound (CirCor) domains, all raw recordings follow a standardized pipeline: (i) Resampling to 16,000 Hz; (ii) Segmentation into fixed 5.0s clips; (iii) Padding / Clipping via repeat padding to handle variable-length audio; and (iv) Normalization of amplitude to [−1.0, 1.0].

For feature-based backbones, CNN6 (Kong et al., 2020a) uses a 128-bin Log-Mel Spectrogram (STFT window 1024, hop 160). CNN14 (Kong et al., 2020a) and BYOL-A use 64-bin Mel-bins, while AST (Gong et al., 2021) and BEATs (Chen et al., 2023) process either raw waveforms or 128-bin spectrograms based on their official pre-trained configurations.

**Optimization and Backbone Configurations.** All models are optimized using the Adam/AdamW optimizer with a Cosine Annealing scheduler and a 10-epoch warm-up. The specific hyperparameter settings for each backbone and task are summarized below:

- **ICBHI Task (4-class)**: CNN6 (Kong et al., 2020a) uses a batch size (BS) of 256, learning rate (LR) of 1e-3, and weight decay (WD) of 1e-4. AST (Gong et al., 2021), BEATs (Chen et al., 2023), FBS (Fraihi et al., 2025), and PAFA (Jeong & Kim, 2025) all use BS=32, LR=5e-5, and WD=1e-6. Training duration is 100 epochs.

- **CirCor Task (3-class)**: Following Niizumi et al. (Niizumi et al., 2024), we train for 50 epochs. CNN14 (Kong et al., 2020a) and BYOL-A (Niizumi et al., 2021) use BS=256 and LR=1e-3. AST (Gong et al., 2021) uses BS=32 and LR=3e-5. M2D (Niizumi et al., 2024) uses BS=32 and LR=2.5e-4. Moreover, SpecAugment (Frequency/Time masking) is applied as: CNN14 (20/200), BYOL-A (20/50), and AST (40/100). M2D and BEATs do not use SpecAugment (0/0).

**Calibration Baseline Settings.** All calibration baselines are implemented using the BEATs backbone on ICBHI:

- **Label Smoothing** (Szegedy et al., 2016). Factor 0.1 applied to Cross-Entropy loss.

- **Mixup** (Zhang et al., 2018). Dirichlet parameter 0.2 for raw signal and label interpolation.

- **Temperature Scaling** (Guo et al., 2017). Scalar T optimized via L-BFGS on the validation set (initial T=1.0).

- **MC Dropout** (Lemay et al., 2022). 5 stochastic forward passes per sample with a 0.1 dropout rate.

- **Deep Ensembles** (Lakshminarayanan et al., 2017). 5 independent models trained with random seeds [1–5]; final prediction is the mean probability distribution.

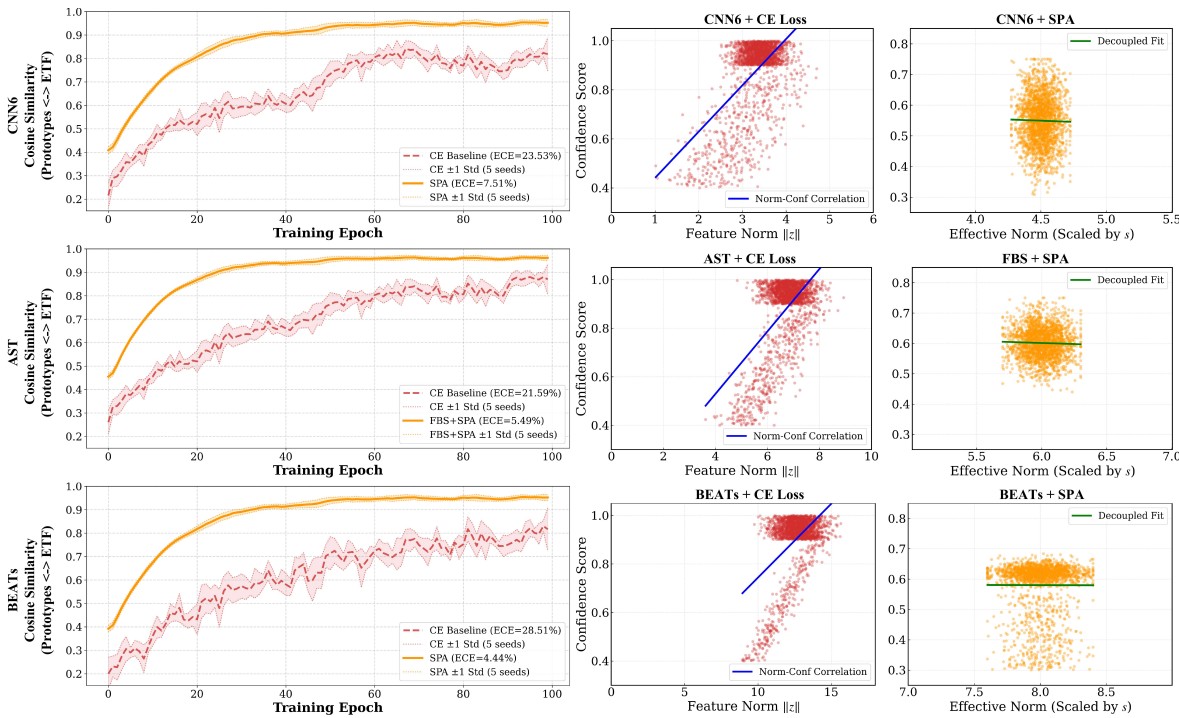

*Figure 5.* Geometric analysis across CNN6, AST, and BEATs backbones on ICBHI dataset. Columns display: (Left) training stability measured by prototype-ETF cosine similarity; (Middle) baseline angular space showing norm-confidence correlation; (Right) SPA angular space showing rectified geometry on the unit hypersphere.

**Practical guidance for new datasets.** From our sensitivity analysis (*Table 5*), we recommend starting with $m = 0.99$, $\tau = 2.0$, $\lambda = 0.5$, and $\omega = 0.5$. For $\lambda$, run a 10-epoch grid over $\{0.1, 0.5, 1.0\}$ (fixing $m, \tau$); select the $\lambda$ with lowest validation ECE provided AS does not drop $> 0.5\%$ from the $\lambda = 0.5$ run. If ECE remains $> 8\%$, increase $\lambda$ to $0.7 - 1.0$; if AS drops $> 1\%$, decrease $\lambda$ to $0.1 - 0.3$. The fusion weight $\omega$ is insensitive: varying $\omega$ between 0.3 and 0.7 on ICBHI changes AS by $\leq 0.5\%$ and ECE by $\leq 0.8\%$, so $\omega = 0.5$ can be fixed without tuning.

### B.3. Additional Visualization Analysis

*Figure 5* validates the geometric benefits of our SPA. As shown in the first column, standard CE training results in significant geometric jitter and wide variance. In contrast, SPA (orange curves) achieves rapid stability via Dynamic Procrustes Alignment. For example, our SPA achieves a stable prototype-ETF similarity of over 0.9 within the first 20–30 epochs, whereas the CE baseline continues to fluctuate within ±1 Std bands until the later stages of training, beyond 80 epochs. The second column highlights that standard backbones suffer from inflated feature norms, e.g., $\|z\|$ up to 15 for BEATs, which causes confidence scores to saturate at 1.0. The third column shows that our SPA constrains features to a learned hypersphere radius $s$, ranging from about 4.5 to 8.0 across backbones, decoupling confidence from magnitude. This shift from unrealistic saturation to a calibrated range centered around 0.6 reduces ECE, confirming that reliability is driven solely by semantic angular alignment.

## C. Workflow and PyTorch Implementation of Our SPA

### C.1. Workflow

*Figure 6* illustrates the core data flow of SPA during training and inference. During training, the backbone outputs feature $z$, which is processed by two branches. The Spherical Branch performs $L_2$ normalization and computes $\text{logits}_s = s \cdot \tilde{W}^\top \tilde{z}$, then updates class prototypes $P$ via momentum (Eq. (4)). The Geometric Branch first adapts $z$ through a learnable adapter to obtain $z_{adp}$, then uses the momentum-updated prototypes $P$ to solve the Orthogonal Procrustes problem via SVD, obtaining the optimal rotation $R^* = \arg\min \|P - RM\|$. The rotated ETF $R^*M$ yields $\text{logits}_g = (R^*M)^\top z_{adp}$. Both logits contribute to the total loss $\mathcal{L}_{\text{total}} = \gamma(\mathcal{L}_{\text{CE}}^s + \mathcal{L}_{\text{CE}}^g) + \lambda \mathcal{L}_{\text{KD}}$. During inference, the two branches produce logits, which are independently softmaxed and fused with equal weight: $P_{\text{final}} = 0.5\,\sigma(\text{logits}_s) + 0.5\,\sigma(\text{logits}_g)$.

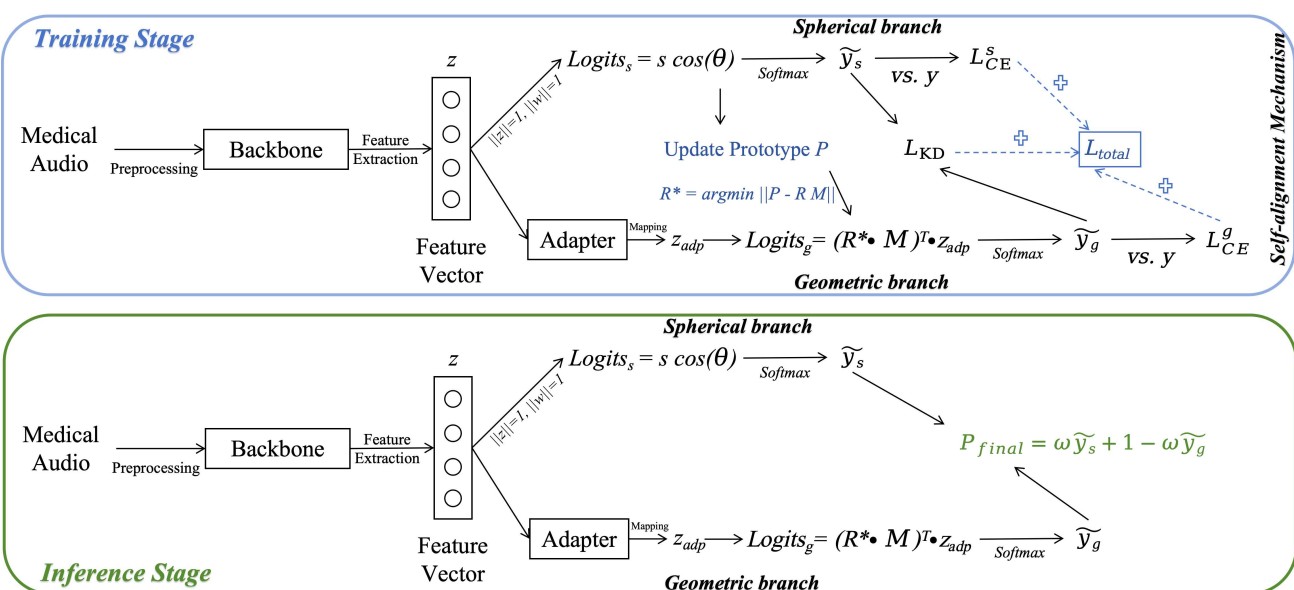

*Figure 6.* Our SPA training and inference data flow.

### C.2. Pytorch implementation

The code can be found at https://github.com/wangying1586/SPA. The class definition of the SPA is as follows:

```python
class SPA(nn.Module):
    """
    Args:
        backbone: Feature extractor; d: feature dim; k: class num.
    Examples:
        >>> model = SPA(backbone, d=2048, k=4).cuda()
        >>> logits = model(x, y) # Jointly aligned and fused logits
    """
    def __init__(self, backbone, d, k, m=0.99, tau=4.0):
        super().__init__()
        self.backbone, self.m, self.tau, self.k = backbone, m, tau, k
        # 1. Spherical Branch: Learnable scale s and normalized weights W
        self.s = nn.Parameter(torch.tensor(16.0))
        self.W = nn.Parameter(torch.randn(k, d))
        self.register_buffer("P", torch.zeros(k, d))  # Prototypes
        # 2. Geometric Branch: Adapter, fixed ETF M, and Orthogonal R
        self.adapter = nn.Linear(d, d)
        C = torch.eye(k) - (1.0 / k)
        etf = torch.cat([C, torch.zeros(d - k, k)], dim=0)
        self.register_buffer("M", F.normalize(etf, p=2, dim=0))
        self.R = nn.Linear(d, d, bias=False)
        geotorch.orthogonal(self.R, "weight")

    def forward(self, x, y=None):
        z = self.backbone(x)
        # --- Spherical Branch ---
        z_n = F.normalize(z, p=2, dim=1)
        logits_s = self.s * F.linear(z_n, F.normalize(self.W, p=2, dim=1))
        if self.training and y is not None:
            # Update prototypes P and perform Dynamic Procrustes Alignment (SVD)
            for i in range(self.k):
                if (y == i).any():
                    self.P[i] = self.m * self.P[i] + (1 - self.m) * z_n[y == i].mean(0)
            self.P.data = F.normalize(self.P, p=2, dim=1)
            U, _, V = torch.svd(self.P.t() @ self.M.t())
            self.R.weight.data = U @ V.t()
        # --- Geometric Branch ---
        z_adp = self.adapter(z)
        logits_g = (z_adp @ (self.R.weight @ self.M)) / self.tau
        # Inference: probability fusion with weight 0.5
        if not self.training:
            prob_s = F.softmax(logits_s, dim=1)
            prob_g = F.softmax(logits_g, dim=1)
            return 0.5 * prob_s + 0.5 * prob_g
        else:
            return(logits_s, logits_g)
```

*Listing 1.* PyTorch implementation of our SPA using torch library

