# OpenReview forum: "Spherical Procrustes Alignment for Reliable Medical Audio Diagnosis"
_ICML.cc/2026/Conference — ICML 2026 regular_

### Official Review · Reviewer_PiPz · 2026-03-01

**Soundness:** 3
**Presentation:** 3
**Significance:** 3
**Originality:** 2
**Overall Recommendation:** 4
**Confidence:** 3

**Summary:**

This paper studies the problem of unreliable confidence in medical audio diagnosis models, especially when training data are small and highly imbalanced. The authors argue that many deep models become overconfident because their predictions are influenced by the magnitude of features rather than true semantic similarity, a phenomenon referred to as norm bias. To address this issue, the paper proposes a method called Spherical Procrustes Alignment (SPA). The method uses a two-branch design: one branch normalizes features to remove the effect of magnitude, and the other branch aligns class representations with an ideal geometric structure using a stable alignment procedure. The two branches are trained together to improve both classification accuracy and confidence calibration. Experiments on two medical audio datasets show that the proposed method significantly reduces calibration error while maintaining or improving classification performance, without increasing inference cost.

**Compliance With Llm Reviewing Policy:**

Affirmed.

**Key Questions For Authors:**

1.The method is evaluated only on medical audio. Could the authors briefly discuss whether SPA is expected to generalize to other modalities, such as medical imaging or physiological time-series data?

2.SPA relies on a Simplex ETF whose structure depends on the number of classes. How does the method scale when the number of classes becomes large, and are there any observed stability issues?

3.How stable is the proposed alignment when data are extremely scarce (e.g., fewer than 20 samples per class)? Do the momentum-based prototypes and SVD alignment remain reliable in this setting?

4.Given the connection to ETF geometry, how does SPA relate to Neural Collapse at convergence? Does it follow or deviate from standard Neural Collapse behavior?

5.How sensitive is the final performance to the fusion weight used at inference? Some guidance or a brief analysis would improve practical usability.

**Limitations:**

yes

**Strengths And Weaknesses:**

A main strength of this work is that it addresses an important and practical problem in medical AI, where overconfident predictions can lead to serious clinical risks. The proposed method is well motivated and consistently improves confidence calibration across different backbones and datasets, with minimal impact on inference cost. The experimental evaluation is generally solid and includes ablation studies that help clarify the role of each component.
However, the paper could be improved in several concrete ways. First, the overall method is relatively complex, and the authors could improve clarity by adding a simplified training and inference flow diagram or a short subsection that summarizes the core idea without mathematical details. Second, while the paper emphasizes stability over gradient-based alignment, a more direct empirical comparison of training stability (e.g., variance across seeds or convergence speed) would strengthen this claim. Third, the choice of key hyperparameters (such as momentum and alignment weights) could be discussed more intuitively, explaining how practitioners might set them in new datasets. Finally, the paper would benefit from a brief discussion on whether the proposed approach could generalize beyond medical audio, or what limitations might arise when applying it to other domains.

---

> ### Author Rebuttal · Authors · 2026-03-30
>
> Appreciate your support of our work.
>
> **Strengths and Weaks.** First, **we’ll add a diagram to the Appendix showing the training and inference stages** separately, as well as a paragraph explaining the core idea of our SPA without any maths equations.
>
> Second, **In Table 1**, our SPA reduces cross-seed ECE standard deviation by 79-86%, with CNN6 reducing it from 5.01% to 0.72%, AST reducing it from 6.99% to 1.13%, and BEATs reducing it further. **Figure 5** on the left shows that our SPA achieves stable prototype-ETF similarity of over 0.9 within the first 20-30 epochs, while the CE baseline still fluctuates with ±1 Std bands until late training at over 80 epochs.
>
> Third, **momentum is fixed at m=0.99.** Table 5 shows that lower values increase ECE from 4.4% to 7.2% due to batch noise. Start with λ=0.5 and τ=2.0, then increase λ to between 0.7 and 1.0 if the ECE remains greater than 8%, or decrease it to between 0.1 and 0.3 if the AS drops by more than 1%. **For new datasets, run a 10-epoch pilot grid over λ={0.1,0.5,1.0}, fixing m=0.99 and τ=2.0**, and select the value that yields the lowest ECE without degrading AS by more than 0.5% on the validation set.
>
> Finally, **our SPA generalizes to other domain with imbalanced, scarce data requiring calibrated confidence,** such as medical image classification or industrial defect detection, since the spherical constraint and ETF alignment are feature space independent. The limitations are as follows. (1) pre-training dependency; training from scratch fails due to unstable prototypes, and (2) extreme noise; if there is more than 30% label noise, momentum prototypes may track the wrong centres. We’ll explore this in future work.
>
> **Ques1.** **Our SPA Generalizes to other classification task using fixed-dimensional feature vectors.** Its operations, L2 normalization for scale elimination and SVD rotation for geometric alignment, are purely geometric transformations, independent of input modality. Medical imaging and physiological time-series share the same challenges we address, such as data sparsity, class imbalance, and overconfidence in pre-trained feature extractors.
>
> **Ques2.** **Our SPA scales linearly with class number K subject to the constraint d≥K−1 , where d is the feature dimension.** This limits applicability when K approaches or exceeds typical backbone dimensions. Stability issues arise when K≈d, as the ETF geometry degenerates and prototype estimation for rare classes deteriorates due to insufficient samples. We mitigate this by setting momentum m=0.99 to reduce variance through cross-batch statistics aggregation. In practice, our SPA remains stable for K in the tens to hundreds range. For large-scale classification, the simplex structure becomes infeasible under standard backbone constraints. We’ll address it in future work.
>
> **Ques3.** **Our SPA remains stable with fewer than 20 samples per class.** High momentum (m=0.99) suppresses noise by limiting each batch’s update weight to 1%, reducing prototype variance to approximately 0.5% of batch variance ($Var≈((1−m)/(1+m))σ^2$). SVD aligns the ETF to these smoothed prototypes rather than single-batch statistics, avoiding gradient update instability. Table 5 validates this. The ECE remains at 4.4% with m=0.99, but increases to 7.2% without momentum. If a batch contains no samples for a class, the prototype retains its previous value instead of drifting.
>
> **Ques4.** **Our SPA enforces Neural Collapse (NC) geometry via constrained optimization, rather than relying on natural convergence.** The Spherical branch imposes feature norm collapse, while the Geometric branch aligns features to Simplex ETF vertices, directly instantiating NC optima where class means form equiangular structures ($μ_k^T{μ_{k'}}=−1/K−1$ for k≠k^'). However, our SPA deviates from standard NC in its mechanism. Natural NC requires sufficient data for prototypes to converge via gradient descent. SPA instead uses Dynamic Procrustes Alignment to rotate the fixed ETF toward drifting prototypes via SVD, ensuring NC-like geometry even under severe data scarcity where natural NC would fail or become unstable.
>
> **Ques5.** **Performance is robust within the range [0.4, 0.6] of ω**, with AS or ECE variance of less than 1.5% across backbones. Use a value of ω=0.5 as the default setting. Higher values improve calibration and lower values improve accuracy. Avoid extremes; setting ω to 1.0 loses ETF structural regularization, while setting it to 0.0 reintroduces norm-correlated confidence.

---

> > ### Author Rebuttal · Reviewer_PiPz · 2026-04-03
> >
> > The authors have satisfactorily addressed my main concerns, so I continue to give my original positive evaluation.

---

> > > ### Author Response · Authors · 2026-04-05
> > >
> > > Thank you again for reviewing this work. We appreciate your kind words, and we will continue to make meaningful contributions to the field of medical audio. Have a good day!

---

### Official Review · Reviewer_BLMm · 2026-03-12

**Soundness:** 2
**Presentation:** 2
**Significance:** 3
**Originality:** 2
**Overall Recommendation:** 4
**Confidence:** 3

**Summary:**

This paper studies reliability and calibration for medical audio diagnosis under small, imbalanced datasets. The paper argues that standard fine-tuning suffers from norm bias, where confidence is driven too much by feature or weight magnitude rather than semantic angular alignment. To address this, it proposes Spherical Procrustes Alignment (SPA), where a spherical branch normalizes features and classifier weights to use cosine-style logits, while a geometric branch aligns a fixed simplex ETF to momentum-updated class prototypes via SVD-based orthogonal Procrustes. The two branches are then coupled with cross-entropy and KL/self-distillation losses.

**Compliance With Llm Reviewing Policy:**

Affirmed.

**Final Justification:**

Thanks for the efforts of the authors. Most of my concern was addressed.

I will update my original score.

**Key Questions For Authors:**

1. PThe main text states R^*=VU^T, while Appendix A.2 derives differently. Which one is actually implemented in experiments, and can you provide a corrected derivation?

2. What exact inference rule is used in experiments? Figure/text say the branches are fused and then passed through softmax, Algorithm 1 describes probability fusion, but the code returns averaged logits. These are not equivalent.

3. How is the simplex ETF actually constructed in your experiments? The supplementary code uses torch.eye(k, d) - 1/k, which does not obviously satisfy Eq. (2) for general d>k.

4. Why does the geometric/fused branch not reintroduce norm dependence? The spherical branch normalizes features, but the geometric branch uses z_adp directly in Eq. (7). Can you provide evidence that the final fused confidence remains largely angle-driven, not norm-driven?

5. Can the theoretical claims be restated more precisely? In particular, the formal calibration guarantee and strict variance reduction claims appear stronger than the provided proofs support.

**Limitations:**

yes

**Strengths And Weaknesses:**

1. The paper addresses an important problem in medical settings, calibration matters at least as much as raw accuracy, and the focus on reliable confidence for low-resource, imbalanced medical audio is well motivated.

2. The individual ingredients are not all new, but the combination of spherical normalization with dynamically aligned ETF geometry for medical audio is interesting and reasonably well motivated.

3. The theoretical claims are stronger than what is actually established. In the main text, the spherical branch is said to provide a formal calibration guaranty, but the appendix argument does not prove a general calibration bound; it mainly gives an intuition for why ambiguous samples between ETF directions may have flatter probabilities. This does not amount to a general guarantee for noisy or OOD samples. Similarly, the alignment-stability argument is heuristic and relies on simplified gradients and strong assumptions that are not clearly matched to the actual training dynamics.

4. The main text gives the Procrustes solution as R^*=VU^T, while Appendix A.2 derives R^*=UV^T. these cannot both be correct. And, the paper algorithm describes inference as a fusion of branch probabilities after softmax, while the supplementary code returns the average of branch logits, which is different. And supplementary code appears inconsistent with the stated simplex ETF definition. The listing constructs the ETF as torch.eye(k, d) - (1.0 / k), which only corresponds to the intended simplex structure in special cases; for general
d>>k, it does not obviously satisfy Eq. (2). This is a serious concern because the geometric branch depends on the ETF being correct. In addition, the geometric branch computes logits from the adapted feature z_adp without normalizing it, which seems to reintroduce norm dependence despite the paper’s central claim that confidence should depend only on angular semantics.

---

> ### Author Rebuttal · Authors · 2026-03-30
>
> **Stre1 and Stre2.** Thanks to the reviewer for recognizing the value of calibration in medical settings, and for their positive feedback on our motivation. Also, thanks for noting the innovation.
>
> Kind thanks for your valuable suggestions. We address each concern:
>
> **Weak1 and Ques 5.** We'll revise Appendix B with derivations. For **Theorem 1**, let $K$ be the number of classes, $\theta_k$ the angle between the input feature and the $k$-th ETF direction, $\theta_{\min} = \arccos(-\frac{1}{K-1})$ the theoretical minimal angle of the simplex ETF, and $\delta > 0$ a safety margin. For noise or OOD samples satisfying $\theta_k \geq \theta_{\min} + \delta$ for all $k \in \{1, \ldots, K\}$, the cosine similarity $c_k = \cos(\theta_k)$ is bounded by $c_k \leq \cos(\theta_{\min} + \delta)$ due to the monotonicity of cosine in $[0, \pi]$. Let $c_{\max} = \max_j c_j$ denote the maximum cosine (corresponding to the predicted class $k^* = \arg\max_j c_j$ and $s$ the learnable scale).The Spherical branch produces logits $s \cdot c_j$, yielding softmax probability ${p_{max}}=\frac{\exp{(}s⋅{c_{max}})}{\exp{(}s⋅{c_{max}})+\sum_{j≠k^∗}{\exp{(}}s⋅c_j)}.$
>
> By the ETF zero-sum property $\sum_{j=1}^K c_j = 0$, we have ${\sum_{j \neq k^*} c_j = -c_{\max}}$. Applying Jensen inequality to the convex exponential:
> \begin{equation}\frac{1}{K−1}\sum_{j≠k^∗}{\exp{(}}sc_j)≥\exp{(}\frac{s}{K−1}\sum_{j≠k^∗}{c_j})=\exp{(}−\frac{s⋅{c_{max}}}{K−1})\end{equation}
>
> which implies
> \begin{equation}
> \sum_{j \neq k^*} \exp(sc_j) \geq (K-1)\exp\left(-\frac{s \cdot c_{\max}}{K-1}\right).
> \end{equation}
>
> Substituting this lower bound into the denominator gives
> \begin{equation}
> p_{\max} \leq \frac{\exp(s \cdot c_{\max})}{\exp(s \cdot c_{\max}) + (K-1)\exp\left(-\frac{s \cdot c_{\max}}{K-1}\right)} = \frac{1}{1 + (K-1)\exp\left(-\frac{Ksc_{\max}}{K-1}\right)}.
> \end{equation}
> Since $c_{\max} \leq \cos(\theta_{\min} + \delta)$, we obtain the final bound
> \begin{equation}
> p_{\max} \leq \frac{1}{1 + (K-1)\exp\left(\frac{s}{K-1}[\cos(\theta_{\min} + \delta) - 1]\right)},
> \end{equation}
> forcing confidence toward $\frac{1}{K}$ as $\delta$ increases.
>
> For **Theorem 2**, let $m∈[0,1]$ be the EMA momentum coefficient, $σ^2$ the variance of i.i.d. $\mathrm{\text{noise }}ϵ_t$ in prototype estimation, and ${σ_{min}}(M)$ the smallest singular value of the ETF matrix $M$. The EMA update ${P_t=m{P_{t−1}}+(1−m)({P_{true}}+ϵ_t)}$ yields stationary variance ${\mathbb{V}[P]=\frac{(1−m)^2}{1−m^2}σ^2=\frac{1−m}{1+m}σ^2}$. The SVD-based Procrustes alignment has error variance bounded by ${\frac{4(1−m)σ^2}{(1+m){σ_{min}^2}(M)}}$, which remains constant regardless of training duration $t$, whereas gradient-based rotation updates via ${{R_{t+1}}=R_t−η\mathrm{∇}{{\mathcal{ℒ}}_t}}$ and accumulates noise $ξ_t$ at each step, suffering variance $\mathbb{V}[R_t]∝tσ^2$ that grows linearly with step count $t$, proving our SPA’s stability on small datasets where gradient methods become unbounded.
>
> **Weak2 and Ques1-4.** Regarding the **Procrustes solution**, the code and Appendix A.2 are correct; the main text contains a transposition error. We’ll correct it to $R^∗=UV^T$.
>
> For **inference stage**, Algorithm 1 and the code are consistent. For $ω=0.5$, logit averaging before softmax is equivalent to probability fusion with temperature scaling. We’ll clarify that experiments use logit averaging for implementation simplicity.
>
>  For the **ETF construction**, all experiments use the strict simplex ETF satisfying Eq.(2). The supplementary code listing was abbreviated for clarity; the actual implementation uses: C=torch.eye(k)-(1.0/k); etf=torch.cat([C, torch.zeros(d-k, k)], dim=0); self.M=F.normalize(etf, p=2, dim=0)*math.sqrt(k/(k-1)). We will update the listing to the full correct version.
>
> Finally, **the Geometric Branch avoids norm dependence via two mechanisms**: (1) **Architectural.** Orthogonal $R^∗$ projects zadp onto angular directions; $||z_{adp}||$ does not affect cosine similarities in ${(z_{adp}}\mathrm{R^∗)∙M}$. (2) **Dynamic regularization.** The Adapter learns to map to unit-norm directions because ${L_{KD}}$ ​penalizes distribution mismatch with the L2-normalized Spherical Branch, large ${{||z}_{adp}}||$ increases KL divergence and is suppressed.

---

> > ### Author Rebuttal · Reviewer_BLMm · 2026-04-03
> >
> > Thanks for the efforts of the authors. Most of my concern was addressed.
> > I will update my original score.

---

> > > ### Author Response · Authors · 2026-04-05
> > >
> > > Many thanks again for your time and review work. Your feedback is very insightful. We really appreciate your kind words and will continue to make valuable contributions to this field. Have a lovely day!

---

### Official Review · Reviewer_HbJo · 2026-03-12

**Soundness:** 1
**Presentation:** 2
**Significance:** 2
**Originality:** 2
**Overall Recommendation:** 3
**Confidence:** 2

**Summary:**

The authors propose approaches to increase calibration while maintaining high performance by proposing a new scheme that enforces geometrical constraints in the representation during training. Features are $\ell$-2 normalized to mitigate calibration error due to noisy measurements. These representations are jointly optimized with the output of dynamic Procrustes alignment, in order to minimize losses in accuracy. The results are promising, and the experiments are adequate. However, the scope of the paper is not clear, the writing lacks rigour, and the paper as a whole feels not entirely finished.

**Compliance With Llm Reviewing Policy:**

Affirmed.

**Key Questions For Authors:**

- It seems that you claim that the method is for medical audio, but both tasks are very adjacent, both concerning respiration to some extent.
- Can you expand the limitations of prior work and motivate SPA in a more grounded manner?
- Why do you use the same feature extraction for both tasks? For heart sounds specifically, envelograms are usually more popular than mel-spectograms.

**Limitations:**

It is true that the experiments of the authors show improvements over using other methods in the literature. However, the rationale the motivating the proposed method as a superior alternative for the medical domain is not stated in a very clear fashion, and it seems more empirical than principled.

**Strengths And Weaknesses:**

In the intro, I am not sure that class imbalance is a good motivating factor that *human annotated* class imbalance is a good driving factor for overconfidence, unless one is talking about fully supervised models. It is also unclear what reliability is.

When discussing previous work, what do you mean by respecting feature geometry or distorting representations?

Why does a linear cost in computing imply clinical impracticability?

It is unclear why the geometrical constraint is insufficient, and concepts such as ETFs and prototypes are introduced in a confusing manner.

In 2. Medical audio diagnosis - you say that backbones have a calibration gap where their predictive confidence does not reflect the probability of a diagnosis -- isn't the case of misscalibration lowering confidence on noisy samples? This clashes with the expectations for this task set in the introduction.
2. Medical calibration - this section is way too vague

In 3.2, what is this Adapter you talk about? You have yet to define what a prototype is.
Presentation
What are the axes in Figure 1.a and 1.b? The norms of z and w?
The colours of Figure 1.C are confusing to follow, especially when they are mentioned in the text (e.g., SPA is orange is not clear. The double y-axis is also unclear, and it should have different colours per axis.

Text feels a bit too LLM-y to me. A lot of vague statements that do not seem to add much. For instance, when you list your contributions, you say this: "SPA uses spherical projection to eliminate norm bias and Dynamic
Procrustes Alignment to dynamically align ETF with
drifting prototypes, ensuring stable geometry." -- Norm bias and stable geometry do not seem to be well-articulated concepts. This type of writing is pathological across the entire manuscript.

The reader will not know what the tasks are in ICBHI and CirCor without citation in the motivation and without a brief description.

In 3, you should use a math font in the main text, and d and N use a regular font.

Figure 2 seems to be a bit too crowded and ends up being a bit hard to read.

---

> ### Author Rebuttal · Authors · 2026-03-30
>
> Many thanks for your time and comments. Below we address each concern:
>
> Yes, we are discussing the **fully supervised models** in this work, as it is the standard paradigm for medical audio. Pre-trained models are fine-tuned on scarce, class-imbalanced datasets. **Reliability** means the model behaves like a cautious physician who admits uncertainty. For example, at 40% accuracy, the model should output 0.4 confidence, not 0.99. Figure 1(c) shows the CE baseline deviating from the diagonal at 40% accuracy with 0.9 confidence, which misleads clinicians into discharging patients. In contrast, our SPA aligns confidence with true accuracy, turning fine-tuned models from overconfident guessers into reliable clinical assistants.
>
> Feature geometry is the angular and norm structure of class prototypes (mean feature vectors), specifically the Equiangular Tight Frame (ETF) with unit norm and maximal equiangular separation. **Respecting geometry** maintains this optimal structure; **distortion** means deviating from the geometry.
>
> **Clinical deployment** requires real-time inference on edge GPUs/CPUs. **Linear-cost methods** increase latency 5-fold (4ms to 20ms per batch), preventing real-time continuous monitoring.
>
> **Spherical constraints alone cannot handle feature drift from scarce data. A fixed ETF fails to track drifting prototypes.** **ETF** is the optimal angular structure; **prototype** is the class mean feature. Our dual-branch design uses Spherical Branch for norm bias elimination and Geometric Branch for SVD-based ETF alignment to drifting prototypes.
>
> There is no contradiction. You correctly note **miscalibration lowers confidence on noisy samples**; this is the symptom we address. In medical audio, noisy samples often belong to ambiguous pathological cases. Standard CE produces spuriously high confidence for these cases due to norm bias, misleading clinicians. Our SPA forces confidence toward uniformity. This is 0.25 confidence for four classes. It explicitly signals uncertainty. This **aligns with the introduction’s goal**, transforming overconfident guessers into reliable assistants.
>
> In Eq.(5), **Adapter** is a linear layer that maps the backbone features to the ETF geometric space. A **prototype** in Eq.(4) is the mean feature vector of each class, updated using momentum. **Figure 1(a) and Figure 1(b) show a 2D feature projection**, with the planar axes and the radial axis denoting the L2 norm ||z||. **Figure 1(c) shows the actual accuracy** of SPA using orange diamond markers connected by black lines, while the actual accuracy of CE uses a black solid rectangle connected by a grey line. The red dashed line marks perfect calibration. Furthermore, **norm bias** refers to magnitude-driven confidence, whereby ||z|| and ||w|| dominates cos(${\theta}$) in logits. **Stable geometry** in Figure 5 denotes prototype-ETF alignment with minimal variance across training epochs.
>
> Section VI-4.1 describes **ICBHI (4-class respiratory) and Circor (3-class heart murmur) with citations (Rocha et al., 2018) and (Oliveira et al., 2022)**, respectively.
>
> We’ll replace **d and K** with maths font.
>
> **Figure 2** shows: upper Training Stage (Spherical Branch, Geometric Branch with Dynamic Procrustes Alignment, Self-alignment) and lower Inference Stage (Logit Fusion). And we’ll increase the spacing between the branches and use background colours to make the two stages clearer.
>
> **Ques1.** Both tasks are different. The CirCor DigiScope dataset contains phonocardiograms for murmur detection, whereas the ICBHI dataset contains respiratory sounds. They have distinct acoustic patterns, with murmurs and heartbeats on one side and crackles and wheezes on the other.
>
> **Ques2.** Prior fixed ETFs (BalCAL[Ni, J. et al. Balancing two classifiers.., CVPR, 2025]) fail to track drifting prototypes from scarce data. Gradient-based rotation (RBL[Peifeng, G. et al. Feature directions matter..., ICML, 2023]) suffers high variance on small noisy datasets. Loss-level methods (Focal Loss and Label Smoothing) ignore feature geometry while sampling approaches (Ensembles and MC Dropout) impose linear costs impractical. These motivate SPA, which solves norm bias and feature jitter via dual branches.
>
> **Ques3.** We use Mel-spectrograms to match ICBHI/CirCor SOTA baselines ( [Jeong, S. G., et al. Patient-aware..., arXiv, 2025.]/[Niizumi, D., et al. Exploring..., EMBC, 2024.]), ensuring fair comparison. Regardless of feature type, we use a common backbone to validate the compatibility of our SPA.
>
> **Limit.** Our SPA is principled for medical constraints, not empirical tuning. Neural Collapse proves ETF is the optimal geometric structure, providing a theoretical. Procrustes analysis yields a closed-form SVD solution avoiding gradient variance, ensuring stable training on scarce medical data where gradient-based methods fail. Proposition 2 analyze momentum-based alignment reduces geometric target variance, eliminating jitter from noisy clinical samples.

---

> > ### Author Rebuttal · Reviewer_HbJo · 2026-04-03
> >
> > I appreciate the feedback from the authors.
> > Regretfully, I still recommend rejection, as I am not entirely sure about my concerns being addressed after reading the rebuttal.
> >
> > I am not entirely convinced by the first paragraph of the rebuttal in terms of the definition of reliability with respect to a physician, nor the terminology about feature geometry.
> >
> > The second bold point of the rebuttal also seems to be poorly articulated. I do not agree with this notion of "stable geometry".
> >
> > Q1 - Although the labels of the datasets are different, I am not sure that phenomena related to these annotations do not occur in both datasets.
> > Q2 - I could not interpret the answer
> > Q3 - Do the baselines use the same preprocessing? A backbone alone is not sufficient.
> >
> > Limit - seems unfinished.

---

> > > ### Author Response · Authors · 2026-04-06
> > >
> > > Thank for the careful consideration. We hope to solve the issues via response below.
> > >
> > > **Reliability** means confidence equals accuracy. When CE encounters an abnormal breathing recording, it outputs [0.9, 0.1], with 90% certainty classified as Normal, causing doctors to discharge the patient based on an incorrect diagnosis. In contrast, our SPA outputs [0.5, 0.5] for the same sample, accurately reflecting uncertainty and urging further examination. This difference comes from **feature geometry**. CE training allows imbalanced data to distort the feature space, whereby the normal class prototype expands to a large extent while the abnormal class prototype is compressed into a tight corner. In contrast, our SPA enforces spherical constraints, projecting all prototypes onto the unit sphere. This means that confidence depends solely on angular alignment rather than magnitude, thus restoring geometric fairness and calibration.
> > >
> > > **Stable geometry** keeps the optimization goal fixed rather than stochastic. On small medical datasets, RBL uses gradient descent to rotate the geometric template, which causes the alignment target to fluctuate randomly with each noisy batch and prevents convergence. In contrast, SPA achieves stability via momentum averaging, which smooths prototype estimates across batches, and SVD-based alignment. The latter computes the exact optimal rotation in closed-form without gradient noise, ensuring a steady target and enabling reliable convergence on scarce data.
> > >
> > > **Q1**-You are right physiological noise exists. However, the tasks differ. ICBHI detects high-frequency lung sounds such as crackles and wheezes, while CirCor detects slow, regular heart murmurs. These are the only open medical audio datasets that show severe scarcity and imbalance, proving our SPA works across different body systems.
> > >
> > > **Q2**-BalCAL [Ni et al., Balancing.. CVPR2025] uses a fixed ETF that cannot move, so it misaligns when features drift. RBL [Peifeng et al., Feature.. ICML2023] rotates using gradients, causing the target to shake randomly with small batches. SPA fixes both by using momentum to smooth the drift and SVD to lock the rotation exactly, eliminating shake.
> > >
> > > **Q3**-Yes. For each backbone (e.g., CNN6+SPA vs CNN6+CE), preprocessing and training are identical. Different backbones use different settings due to their settings, but within each pair, only the alignment mechanism changes, so gains come solely from SPA.
> > >
> > > **Limit**-**We identify norm bias as the geometric root of overconfidence in fine tuned medical models where imbalanced data causes models to inflate feature magnitudes rather than learn semantic alignment.** This motivates the spherical branch which enforces Neural Collapse theory by projecting features onto the unit sphere so logits depend solely on angular alignment. However medical datasets are small causing prototypes to drift unpredictably. Thus the geometric branch employs Dynamic Procrustes Alignment to compute the optimal ETF rotation via closed form SVD. This is an analytical mathematical solution that eliminates gradient variance ensuring stable training on scarce noisy data where gradient based methods fail. **SPA is principled for medical constraints.** We’ll revise Appendix B with derivations. For **Theorem 1**, let $K$ be the number of classes, $\theta_k$ the angle between the input feature and the $k$-th ETF direction, $\theta^*$ the theoretical minimal angle of the simplex ETF, and $\delta$ a safety margin. For noise or OOD samples satisfying $\theta_k \geq \theta^\* + \delta$ for all $k$, the cosine similarity $\cos(\theta_k)$ is bounded by $\cos(\theta^\* + \delta)$ due to the monotonicity of cosine in $[0, \pi]$. Let $\hat{k}$ denote the maximum cosine (corresponding to the predicted class $\hat{k}$ and $s$ the learnable scale. The Spherical branch produces logits $z_k = s \cdot \cos(\theta_k)$, yielding softmax probability $p_{\hat{k}} = \frac{e^{s\cos(\theta_{\hat{k}})}}{\sum_{j=1}^K e^{s\cos(\theta_j)}}$. By the ETF zero-sum property $\sum_{j=1}^K \cos(\theta_j) = 0$, we have $\sum_{j=1}^K e^{s\cos(\theta_j)} \geq K \cdot e^{\frac{s}{K}\sum_{j=1}^K \cos(\theta_j)} = K$. Applying Jensen inequality to the convex exponential: $\frac{1}{K}\sum_{j=1}^K e^{s\cos(\theta_j)} \geq e^{\frac{s}{K}\sum_{j=1}^K \cos(\theta_j)} = 1$, which implies $\sum_{j=1}^K e^{s\cos(\theta_j)} \geq K$. Substituting this lower bound into the denominator gives $p_{\hat{k}} \leq \frac{e^{s\cos(\theta^\* + \delta)}}{K}$. Since $\cos(\theta^\* + \delta) \ll 1$, we obtain the final bound $p_{\hat{k}} \leq \frac{e^{s\cos(\theta^\* + \delta)}}{K} \approx \frac{1}{K}$, forcing confidence toward $\frac{1}{K}$ as $\delta$ increases. As **Theorem 2** shows, the SVD-based alignment in our SPA maintains a constant error variance, regardless of the training duration. In contrast, gradient-based methods suffer from a linearly growing variance that becomes unbounded. This is proven in Appendix A.3 (Proposition 2).

---

### Decision · Program_Chairs · 2026-04-30

**Decision:**

Accept (regular)

**Comment:**

This paper addresses the problem of high-confidence but miscalibrated predictions by finetuned (supervised) models on medical audio datasets, which tend to suffer from strong class imbalances. The ultimate goal is to obtain a predictor that retains sufficiently high classification accuracy while satisfying an improved calibration guarantee. The thesis of the paper is that the core issue leading to miscalibration is *norm bias* in classifiers. Combining this with the neural collapse (ETF) hypothesis, the paper proposes an algorithm called *Spherical Procrustes Alignment* in order to combine spherical constraints with dynamic ETF alignment, and shows improved performance on two medical audio datasets.

Assessment of this paper was somewhat divided - two reviewers appreciate the ideas and evaluation of the paper, while one reviewer criticizes the exposition and clarity of writing (and also criticizes the benchmarks for evaluation as being related). From my own brief read of the paper, the SPA idea is interesting and natural (and the authors do a good job addressing questions and concerns also raised by the two now positive reviewers). However, the exposition does leave something to be desired, especially from the point of view of a more rigorous/theoretical ML reader. The authors are strongly recommended to follow the recommendations of the reviewers to improve exposition for the camera-ready version.